# Dynamic readout of the Hh gradient in the *Drosophila* wing disc reveals pattern-specific tradeoffs between robustness and precision

**Rosalío Reyes[1,2], Arthur D Lander[3], Marcos Nahmad[1]***

[1]Department of Physiology, Biophysics, and Neurosciences; Center for Research and Advanced Studies of the National Polytechnic Institute (Cinvestav), Mexico City, Mexico; [2]Interdisciplinary Polytechnic Unit of Biotechnology of the National Polytechnic Institute, Mexico City, Mexico; [3]Department of Developmental and Cell Biology and Center for Complex Biological Systems, University of California, Irvine, Irvine, United States

**Abstract** Understanding the principles underlying the design of robust, yet flexible patterning systems is a key problem in developmental biology. In the *Drosophila* wing, Hedgehog (Hh) signaling determines patterning outputs using dynamical properties of the Hh gradient. In particular, the pattern of *collier* (*col*) is established by the steady-state Hh gradient, whereas the pattern of *decapentaplegic* (*dpp*), is established by a transient gradient of Hh known as the Hh overshoot. Here, we use mathematical modeling to suggest that this dynamical interpretation of the Hh gradient results in specific robustness and precision properties. For instance, the location of the anterior border of *col*, which is subject to self-enhanced ligand degradation is more robustly specified than that of *dpp* to changes in morphogen dosage, and we provide experimental evidence of this prediction. However, the anterior border of *dpp* expression pattern, which is established by the overshoot gradient is much more precise to what would be expected by the steady-state gradient. Therefore, the dynamical interpretation of Hh signaling offers tradeoffs between robustness and precision to establish tunable patterning properties in a target-specific manner.

**\*For correspondence:**
mnahmad@fisio.cinvestav.mx

## Editor's evaluation

This study presents a valuable finding on the precision conferred by dynamical interpretation of morphogen gradients. The evidence supporting the claims of the authors is convincing, with compelling theoretical analysis and solid experimental data. The authors have adequately addressed most concerns raised and so the work will be of considerable interest to the developmental biology and developmental systems biology communities.

## Introduction

Developmental patterning must be robust to variety of genetic and environmental perturbations in order to ensure a reproducible and functional body plan. Since patterns of gene expression are often specified by morphogen gradients, there has been considerable interest in understanding how these gradients reliably establish positional boundaries (*Neumann and Cohen, 1997*; *Gurdon and Bourillot, 2001*; *Lander, 2007*; *Claret et al., 2007*; *Ibañes and Izpisúa Belmonte, 2008*; *Rogers and Schier, 2011*; *Li et al., 2018*; *Stapornwongkul et al., 2018*). This reliability depends on the

robustness of pattern specification with respect to different perturbations, as well as the precision or sharpness of pattern boundaries. Several theoretical studies have investigated the properties in which patterning robustness is ensured (*Eldar et al., 2003*; *Bergmann et al., 2007*; *Lander et al., 2009*; *Adelmann et al., 2023*). These studies are generally based solely on steady-state morphogen profiles and therefore, robustness applies equally to all patterning targets. As a result, steady-state morphogen gradients cannot tune these patterning properties in a target-specific manner. The *Drosophila* wing imaginal disc has become a useful system to study the mechanisms of morphogen formation and interpretation and offers testable patterning outputs in terms of both robustness and precision in the adult wing (*Hartl and Scott, 2014*; *Restrepo et al., 2014*; *Chen and Zou, 2019*). Along the anterior–posterior (AP) axis, the *Drosophila* wing is patterned by the Hedgehog (Hh) and Decapentaplegic (Dpp) morphogen gradients that determine the position of the longitudinal veins L2–L5 (*Blair, 2007*). Hh is produced in cells of the posterior compartment during the third larval instar and forms a short-range signaling gradient into the anterior compartment (*Tabata and Kornberg, 1994*). The Hh gradient organizes AP patterning of the wing both directly and indirectly; it defines adult patterning outcomes, such as the expression of the transcription factor *knot* or *collier (col)* which sets the distance between the longitudinal veins L3 and L4 (*Vervoort et al., 1999*; *Anonymous, 2000*); and the expression of *decapentaplegic* (*dpp*) in a domain broader than *col* (*Basler and Struhl, 1994*; *Vervoort, 2000*). While *dpp* does not have a direct patterning output in the adult wing, Dpp then acts as a long-range morphogen to globally coordinate patterning and growth along the AP axis (*Affolter and Basler, 2007*).

Contrary to other signaling pathways in which a ligand activates a signaling cascade by binding to its receptor, Hh signaling is activated by removing the receptor Patched (Ptc) from the plasma membrane, a process that is promoted by Hh binding and endocytosis (*Torroja et al., 2005*). This suggests that Hh signaling activity solely depends on the number of unbound Ptc receptors. However, a study suggested that the levels of Hh-bound Ptc can titrate the inhibitory effects of unbound Ptc and proposed that Hh signaling activity is more accurately represented by the ratio of bound to unbound Ptc receptor (*Casali and Struhl, 2004*). Importantly, an evolutionary conserved feature of the Hh signaling pathway is that *ptc* is itself a target of the signal. Since Ptc expression attenuates the dispersion and strength of signaling activity, Hh-dependent Ptc upregulation acts as a negative feedback that self-limits the range of the gradient (*Chen and Struhl, 1996*; *Briscoe et al., 2001*). This feedback property of Hh signaling results in self-enhanced ligand degradation which makes a narrower, but more robust gradient to perturbations in ligand dosage (*Eldar et al., 2003*; *Lander et al., 2009*).

Hh-dependent Ptc upregulation also provides an alternative interpretation of positional information, in which instead of using multiple concentration thresholds of the Hh steady state as in the classical morphogen model, patterning is established by interpreting positional information in a temporal manner using a single-threshold signaling range defined by a transient and the steady-state gradients (*Nahmad and Stathopoulos, 2009*). In particular, the boundary of *dpp* is established by an extended pre-steady-state gradient, known as the *overshoot*, while the anterior border of *col* is established by the steady-state gradient. Since the overshoot occurs prior to Hh-dependent Ptc upregulation, *dpp* should not exhibit the robustness property offered by the self-enhanced ligand degradation mechanism, but this has not yet been documented experimentally.

A study by Irons et al. compared the width of *col* expression in the wing disc as well as the L3–L4 intervein distance in adult wings of *hh* heterozygous and wild-type animals and found that they are not statistically different, supporting that some robustness to Hh dosage is exhibited by the system (*Irons et al., 2010*). Furthermore, Hatori et al. showed that the widths of *col* or *ptc* patterns do not significantly change in discs with 1, 2, 3, or 4 *hh* gene copies (*Hatori et al., 2021*). However, it remains unclear if the same robustness is exhibited by *dpp* which depends on the dynamics of the Hh gradient. By using mathematical modeling, here we show that when patterns are established by steady-state models of patterning all target genes exhibit the same robustness with respect to changes in morphogen production, in agreement with prior theoretical work (*Eldar et al., 2003*). However, when the Hh gradient is interpreted dynamically through the overshoot model (*Nahmad and Stathopoulos, 2009*), robustness to *hh* dosage becomes target specific. In particular, the specification of the anterior border of *col* is more robust than that of *dpp*, since the latter is independent of Hh-dependent Ptc upregulation. In contrast, we show that the anterior border of *dpp* model under the overshoot model offers increased precision, relative to what would be expected in the

steady state only patterning model. Taken together, our work shows that the overshoot model of Hh signaling enables tunable robustness and precision properties in a target-specific manner. We discuss implications of this dynamic patterning model in the context of balancing reliability and flexibility during developmental patterning.

## Results

## Steady-state interpretation of morphogen gradients predicts identical robustness to morphogen dosage for all targets

Prior work on morphogen robustness has relied on quantifying displacements of the overall gradient shape (*Gurdon and Bourillot, 2001*; *Tabata and Takei, 2004*) or a single-threshold location of a gradient (*Eldar et al., 2003*). Robustness can be measured by computing the displacement ($\Delta x$) of the pattern boundary defined by a given morphogen threshold concentration, $T$, as result of a specific perturbation:

$$\Delta x = |x(T) - \tilde{x}(T)|, \tag{1}$$

where $x(T)$ and $\tilde{x}(T)$ are the positions defined by the concentration threshold $T$ of the unperturbed and perturbed morphogen gradients, respectively. Since *Equation 1* is an absolute measure of robustness, in practice, perfect robustness occurs when $\Delta x$ is less than the diameter of a single cell.

To investigate robustness of different target genes, we first analyze robustness predicted by classical morphogen models, that is, in which territories are defined by different thresholds of the steady-state gradient. As a starting model, we consider a free-diffusion, linear-degradation model at the steady state:

$$\frac{d^2 M}{dx^2} - \frac{1}{\lambda^2} M = 0, \tag{2}$$

where $M$ is the concentration of the morphogen and $\lambda^2$ is the square of the characteristic gradient length, defined by the ratio between the diffusion coefficient and the degradation rate of the ligand $M$, subject to the following boundary conditions:

$$\begin{aligned} B.C.1. \quad & M(0) = M_0, \\ B.C.2. \quad & \lim_{x \to \infty} M(x) = 0. \end{aligned} \tag{3}$$

In this case, a perturbation in the morphogen source, $M_0 \to \tilde{M}_0$, results in a uniform displacement of the gradient which is given by $\Delta x = \lambda \ln(\tilde{M}_0/M_0)$ (*Eldar et al., 2003*), showing that patterns established by different thresholds exhibit the same response to this perturbation. This occurs because the solution of the perturbed problem is just a constant shift of the morphogen profile (*Figure 1b–d*).

We then considered a very simple model of Hh signaling in the *Drosophila* wing. Since the expression of Ptc, the Hh receptor, is upregulated by Hh signaling and contributes to Hh degradation by binding the Hh ligand, we considered a model in which ligand degradation has different values within and beyond a presumptive Ptc expression domain:

$$\frac{\partial Hh}{\partial t} = D \frac{\partial^2 Hh}{\partial x^2} + \theta(x)\alpha_{Hh} - \beta(x,t)Hh \tag{4}$$

where $\theta(x)\alpha_{Hh}$ represents the source of Hh in the posterior compartment of the wing disc (i.e., $\theta(x)$ is equal to 1 or 0, depending on whether $x$ is a location in the posterior [$x < 0$] or anterior compartment [$x > 0$], respectively), and

$$\beta(x,t) = \gamma_{Hh\_Ptc} Ptc + \beta_{Hh}, \tag{5}$$

where $\gamma_{Hh\_Ptc}$ is the mass action constant for *Hh_Ptc* binding. At the steady state, we expect that *Ptc* forms a uniform expression pattern over a stripe of anterior cells abutting the AP border (referred as $Ptc_{ss}$) and away from the stripe, *Ptc* is expressed at basal levels, $Ptc_0$. Then, at the steady state $\beta_{\text{steady-state}}(x)$ can be modeled as the step function

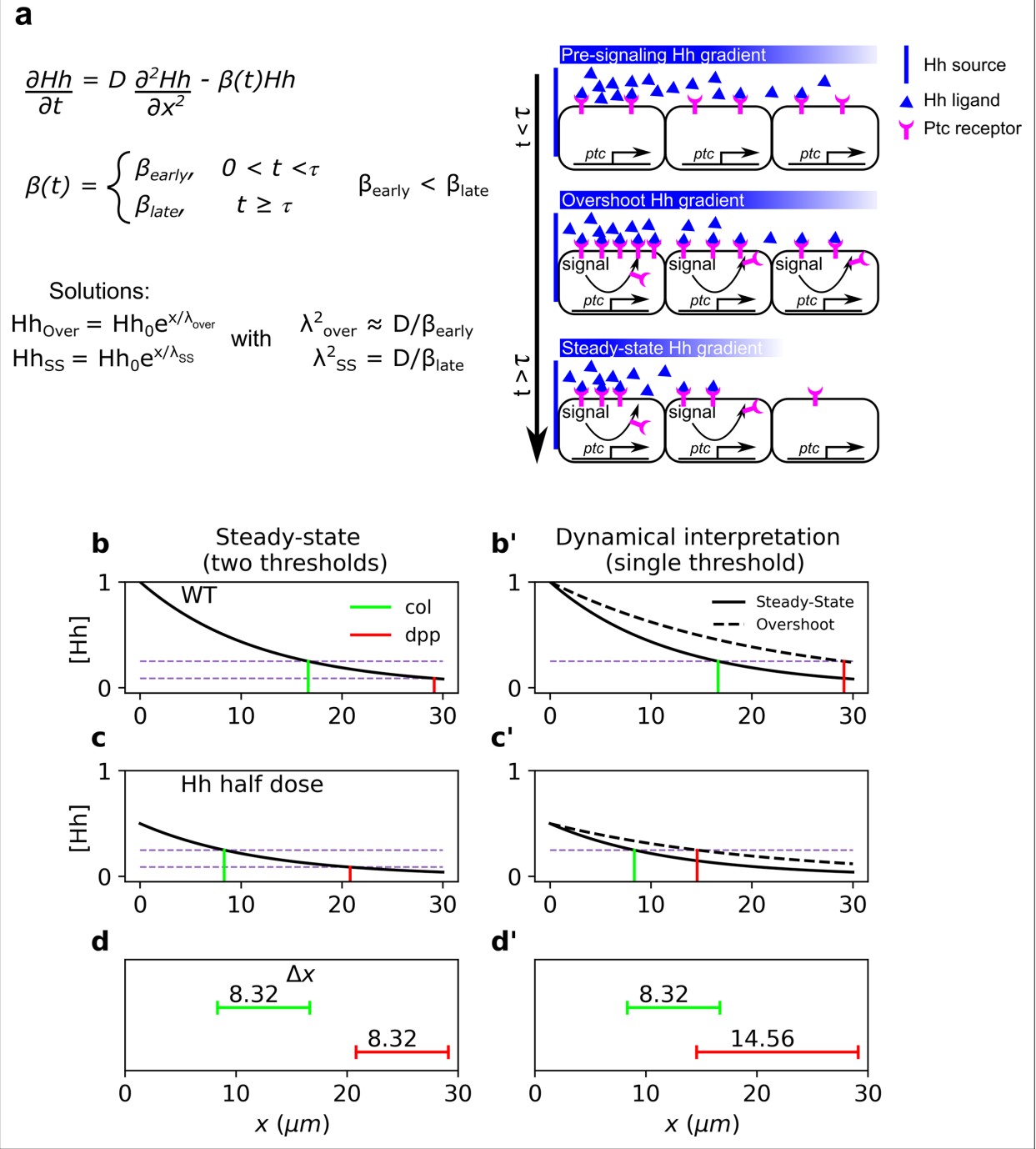

**Figure 1.** Dynamical interpretation model predicts differential robustness when morphogen dosage is reduced to half. (**a**) Simple model of Hh signaling using a time-dependent step-wise degradation function. Diagrams displays a pre steady-state gradient that then retracts upon Hh-dependent *ptc* upregulation, resulting in a narrower gradient. (**b, c**) Plots of the analytical solution for the model in a using full ($Hh(x = 0) = 1$); (**b, b'**) or half ($Hh(x = 0) = 0.5$); (**c, c'**) Hh dosage. (**d, d'**) Displacements upon the above perturbation for the steady-state model with two thresholds (dotted horizontal lines corresponding to the locations of *col* and *dpp*) d; and for the dynamical interpretation model with a single-threshold readout (single dotted horizontal line) using the overshoot vs. the steady-state gradient predicts different shifts d'. The parameter values used for these plots are: $\lambda_{over} = 21\mu m$, $\lambda_{SS} = 12\mu m$ which approximately correspond to the anterior border positions of *col* and *dpp*, respectively. The color coding of *dpp* in red and *col* in green, will be used in the rest of the article.

The online version of this article includes the following source code for figure 1:

**Source code 1.** Code to generate *Figure 1*.

$$\beta_{\text{steady-state}}(x) = \begin{cases} \gamma_{Hh\_Ptc}Ptc_{ss} + \beta_{Hh}, & 0 < x < b, \\ \gamma_{Hh\_Ptc}Ptc_0 + \beta_{Hh}, & \text{otherwise}, \end{cases} \tag{6}$$

where $b$ is the width of the Ptc stripe. For $x > 0$, the steady-state solution of *Equation 4* is given by

$$Hh(x) = \begin{cases} Hh_{\text{stripe}}(x) = Ae^{x/\lambda_2} + Be^{-x/\lambda_2}, & 0 < x < b \\ Hh_{\text{beyondPtc}}(x) = Ce^{-x/\lambda_1}, & x \geq b. \end{cases} \tag{7}$$

where $\lambda_1$ and $\lambda_2$ are the morphogens characteristic lengths within and beyond the Ptc stripe, and $A$, $B$, and $C$ are constants determined by the boundary conditions. Upon a perturbation $\alpha_{Hh} \to \tilde{\alpha}_{Hh}$, perturbed Hh concentrations are given by:

$$\tilde{Hh}_{\text{stripe}}(x) = Ae^{\left[x + \frac{1}{\lambda_2}\ln\left(\frac{\tilde{\alpha}_{Hh}}{\alpha_{Hh}}\right)\right]/\lambda_2} + Be^{-\left[x - \frac{1}{\lambda_2}\ln\left(\frac{\tilde{\alpha}_{Hh}}{\alpha_{Hh}}\right)\right]/\lambda_2}, \tag{8}$$

and

$$\tilde{Hh}_{\text{beyondPtc}}(x) = Ce^{-\left[x - \frac{1}{\lambda_1}\ln\left(\frac{\tilde{\alpha}_{Hh}}{\alpha_{Hh}}\right)\right]/\lambda_1}. \tag{9}$$

Note that once again, all territories defined by $Hh_{\text{beyondPtc}}$ are shifted by the same amount, $\lambda_1 \ln\left(\frac{\tilde{\alpha}_{Hh}}{\alpha_{Hh}}\right)$, upon variations in $\alpha_{Hh}$. Therefore, any two target genes whose borders are defined by different concentration thresholds will exhibit the same robustness response.

## Dynamic models of Hh signaling using a single threshold for different targets predict differential robustness

Previous work showed that Hh signaling in the *Drosophila* wing disc the anterior border of the Hh targets *dpp* and *col* are established by a single threshold at two time points during the formation of the Hh gradient; namely, at the overshoot and the steady state, respectively (*Nahmad and Stathopoulos, 2009*). To consider this dynamical patterning mechanism, we analyzed a simplified model which takes into accountx the temporal upregulation of *ptc* as a time-dependent switch function (*Figure 1a*). Following the overshoot model in *Nahmad and Stathopoulos, 2009*, we defined the overshoot gradient as the transient profile of maximum range. Since the timescale of Hh diffusion is much faster than the timescale of Ptc upregulation, we will assume that the Hh gradient reaches a pre-steady state with the first degradation rate, $\beta_{early}$, where the anterior border of *dpp* is approximately defined and then the real steady state with the second degradation rate, $\beta_{late}$ (*Figure 1b'*). Under this simple model of Hh signaling, the shift in patterning borders defined by the overshoot (i.e., *dpp*) and the displacement at the steady state are related by the following simple equation:

$$\Delta x_{SS} = \frac{\lambda_{SS}}{\lambda_{over}} \Delta x_{over}, \tag{10}$$

where $\lambda_{over}$ and $\lambda_{SS}$ are the morphogen characteristic lengths before and after *ptc* upregulation, respectively (see *Figure 1a*). Since $\beta_{early} < \beta_{late}$, then $\lambda_{SS} < \lambda_{over}$ and $\Delta x_{SS} < \Delta x_{over}$, that is, overshoot-dependent targets are less robust than those established by the steady-state gradient. Then, in contrast to the steady-state model (*Figure 1b–d*), the overshoot model predicts differences in target gene displacement upon perturbation of morphogen dosages (*Figure 1b'–d'*), that is, robustness is target dependent, with higher robustness predicted for *col* patterning due to self-enhanced ligand degradation, than for *dpp* patterning (*Figure 1d, d'*). The ratio $\lambda_{SS}/\lambda_{over}$ in *Equation 10* may be written in terms of the kinetic parameters of Hh signaling (see *Equation 6*):

$$\Delta x_{SS} = \frac{\beta_{over}}{\beta_{SS}} \Delta_{over} = \frac{\gamma_{Hh\_Ptc}Ptc_{SS} + \beta_{Hh}}{\gamma_{Hh\_Ptc}Ptc_{over} + \beta_{Hh}} \Delta x_{over} \approx \frac{Ptc_{SS}}{Ptc_{over}} \Delta x_{over}. \tag{11}$$

The last approximation, which assumes that Ptc-dependent Hh degradation is much faster than other means of Hh degradation, provides an estimate of the difference in robustess for overshoot

and steady-state targets as a function of Ptc levels. Note that in *Equation 11*, the difference in $\Delta x$ between the steady state and overshoot model is independent of the specific threshold at which the Hh gradient establishes positional information. Thus, this equation provides a way to experimentally relate pattern robustness to actual patterning outputs in the system, such as Ptc expression levels (see Discusion).

We then asked if these results also hold in a more explicit model of the Hh pathway (*Nahmad and Stathopoulos, 2009*):

$$\frac{\partial Hh}{\partial t} = D\frac{\partial^2 Hh}{\partial x^2} + S^+(x)\alpha_{Hh} - \gamma_{Hh\_Ptc}Hh \times Ptc - \beta_{Hh}Hh, \tag{12}$$

$$\frac{\partial ptc}{\partial t} = S^-(x)\alpha_{ptc0} + \frac{\alpha_{ptc}Signal^m}{k_{ptc}^m + Signal^m} - \beta_{ptc}ptc, \tag{13}$$

$$\frac{\partial Ptc}{\partial t} = \mu_{Ptc}ptc - \gamma_{Hh\_Ptc}Hh \times Ptc - \beta_{Ptc}Ptc, \tag{14}$$

$$\frac{\partial Hh\_Ptc}{\partial t} = \gamma_{Hh\_Ptc}Hh \times Ptc - \beta_{Hh\_Ptc}Hh\_Ptc, \tag{15}$$

$$\frac{\partial Signal}{\partial t} = \frac{S^-(x)\alpha_{Signal}\left(\frac{Hh\_Ptc}{Ptc}\right)^n}{k_{Signal}^n + \left(\frac{Hh\_Ptc}{Ptc}\right)^n} - \beta_{Signal}Signal, \tag{16}$$

where *Hh*, *ptc*, *Ptc*, and *Hh_Ptc* are the concentrations of Hh, *ptc* (mRNA), Ptc (protein), and the Hh-Ptc complex, respectively. The coefficients $\alpha$, $\beta$, $\gamma$, and $\mu$ represent the rates of synthesis, degradation, complex formation, and translation, respectively (see *Figure 2-source data 6*). We used a system of coordinates centered on the AP boundary with the anterior compartment on the negative side. $S^+(x)$ [alternatively, $S^-(x)$] is a step function of the form $S^+(x) = 1$ if $x > 0$ (alternatively, $S^-(x) = 1$ if $x < 0$) and zero otherwise. *Signal* represents the intracellular response of Hh signaling activity that activates target gene expression. The system of *Equations 12–16* is subject to the following boundary and initial conditions:

$$\begin{aligned}
&\text{I.C. 1} \quad ptc(x, 0) = S^-(x)\frac{\alpha_{ptc0}}{\beta_{ptc}}, \\
&\text{I.C. 2} \quad Ptc(x, 0) = \frac{\mu_{Ptc}}{\beta_{Ptc}}ptc(x, 0) = S^-(x)\frac{\alpha_{ptc0}\mu_{Ptc}}{\beta_{ptc}\beta_{Ptc}}, \\
&\text{B. C.} \quad \left.\frac{\partial Hh}{\partial x}\right|_{x=-100} = \left.\frac{\partial Hh}{\partial x}\right|_{x=100} = 0.
\end{aligned} \tag{17}$$

We solved *Equations 12–16* numerically and computed $\Delta x$ (as in *Equation 1*) for the overshoot and steady-state *Signal* gradients upon a range of perturbations of the wild-type Hh production rate, $\alpha_{Hh_0}$ (*Figure 2a*). In agreement with our previous result (*Figure 1*), we found that the steady-state outputs are more robust than the overshoot outputs (*Figure 2a*). Moreover, this result holds independently of the specific choice of model parameters (*Figure 2b*). We conclude that higher robustness is predicted for targets specified by the steady-state gradient (*col*), with respect to those specified by the overshoot profile (*dpp*).

## Robustness of steady-state outputs depends on Hh-dependent Ptc regulation

Since previous work suggests that Hh-dependent *ptc* upregulation determines the range of the signal (*Chen and Struhl, 1996*), we wanted to confirm that Hh-dependent *ptc* regulation is responsible for the difference in robustness of Hh outputs. We perturbed the *ptc* production rate, $\alpha_{ptc}$, and noticed that $\Delta x$ computed using the steady-state *Signal* profile is clearly reduced, but has little effect when computed with the overshoot *Signal* function (green vs. red dots in *Figure 2b*). Once again, this result is largely independent of the choice of parameters since robustness always improves compared to the case when $\alpha_{ptc} = 0$ (*Figure 2b'*). Therefore, we suggest that Hh-dependent Ptc upregulation is responsible for differential robustness in this system by making steady-state outputs more robust with respect to overshoot-defined outputs.

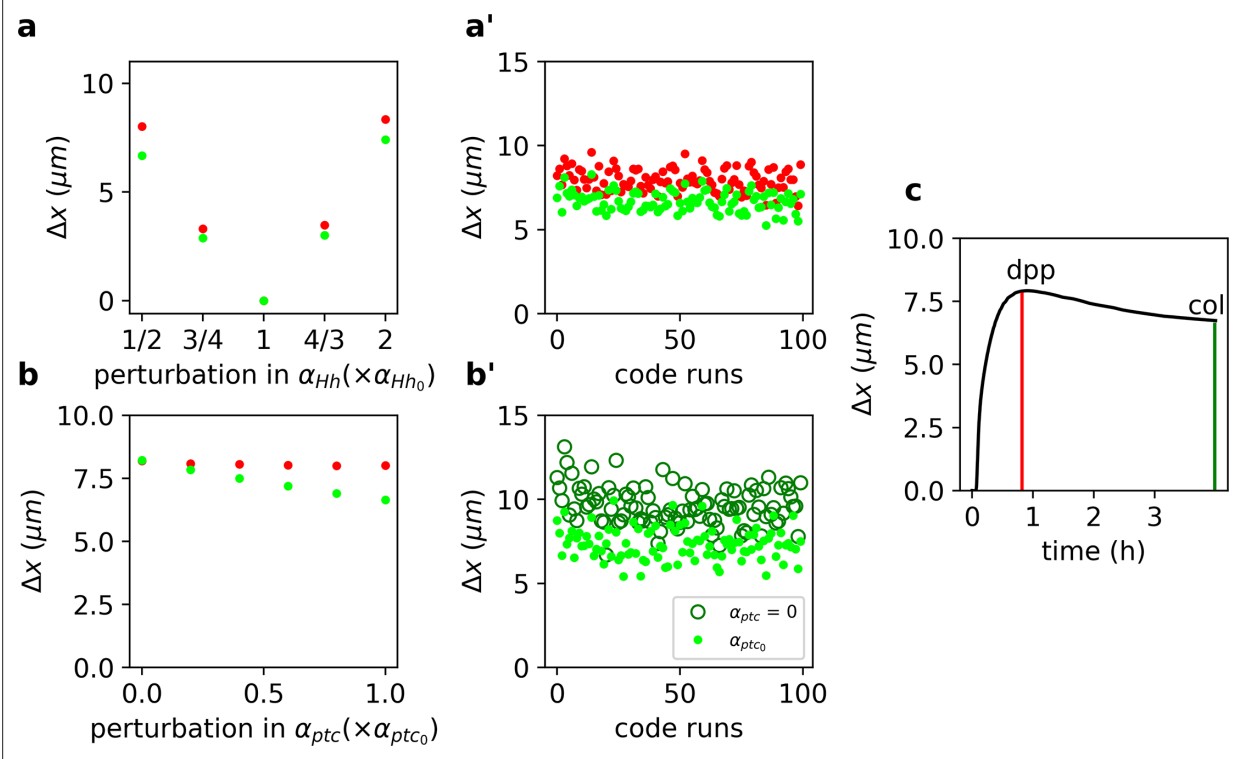

**Figure 2.** Target-specific robustness still holds in an explicit model of Hh signaling and it is dependent on Hh-dependent Ptc upregulation. (**a**) $\Delta x$ (defined as in *Equation 1*, but for the *Signal* function, see Materials and Methods) for overshoot (red) vs. steady-state (green) outputs upon different perturbations in $\alpha_{Hh}$ using the values of the parameters reported in *Nahmad and Stathopoulos, 2009* (*Figure 2—source code 2 and 3* and *Figure 2—source data 1, 2, and 6*). (**a'**) $\Delta x$ defined and color coded as in a, for different combinations of parameter runs, when all parameters (other than $\alpha_{Hh}$) are varied through a random normal distribution around the mean value with a standard deviation of 10% of the mean value (*Figure 2—source data 2*). (**b**) Same as a, but for perturbations in $\alpha_{ptc}$ (*Figure 2—source data 3*). (**b'**) Comparison of $\Delta x$ for different parameters runs as in a' for steady-state outputs (light green dots) and when $\alpha_{ptc=0}$ (dark green empty circles; *Figure 2—source data 4*). (**c**) $\Delta x$ defined as in a, computed for the *Signal* gradient over time (*Figure 2—source data 5*). Red and green vertical lines indicate the overshoot and steady-state values corresponding to the anterior borders of *dpp* and *col*, respectively (*Figure 2—source code 1*).

The online version of this article includes the following source data for figure 2:

**Source code 1.** Code to generate *Figure 2*.

**Source code 2.** Code to solve steady-state solution of *Equation 18*.

**Source code 3.** Code to solve transient solution of *Equations 12–17*.

**Source data 1.** Raw data to generate *Figure 2a*.

**Source data 2.** Raw data to generate *Figure 2a'*.

**Source data 3.** Raw data to generate *Figure 2b*.

**Source data 4.** Raw data to generate *Figure 2b'*.

**Source data 5.** Raw data to generate *Figure 2c*.

**Source data 6.** Parameters used to solve *Equations 12–17* (same values as in *Nahmad and Stathopoulos, 2009*).

Prior theoretical work suggests that when positional information is established before the steady state, it enhances robustness (*Bergmann et al., 2007*). This idea appears to contradicts our finding that overshoot-dependent patterning (which occurs prior to steady state) is less robust than steady-state-dependent patterning (*Figure 2a, b*). In order to understand the relative robustness of pre-steady-state gradients, we computed $\Delta x$, upon perturbations of $\alpha_{Hh}$ as a function of time in our model of Hh signaling. We found that early transient states exhibit the smallest $\Delta x$ and therefore are the gradients that drive the more robust outputs, although they have a very limited range (*Figure 2c*), in agreement with the study of *Bergmann et al., 2007*. Then, $\Delta x$ increases as the gradient approaches

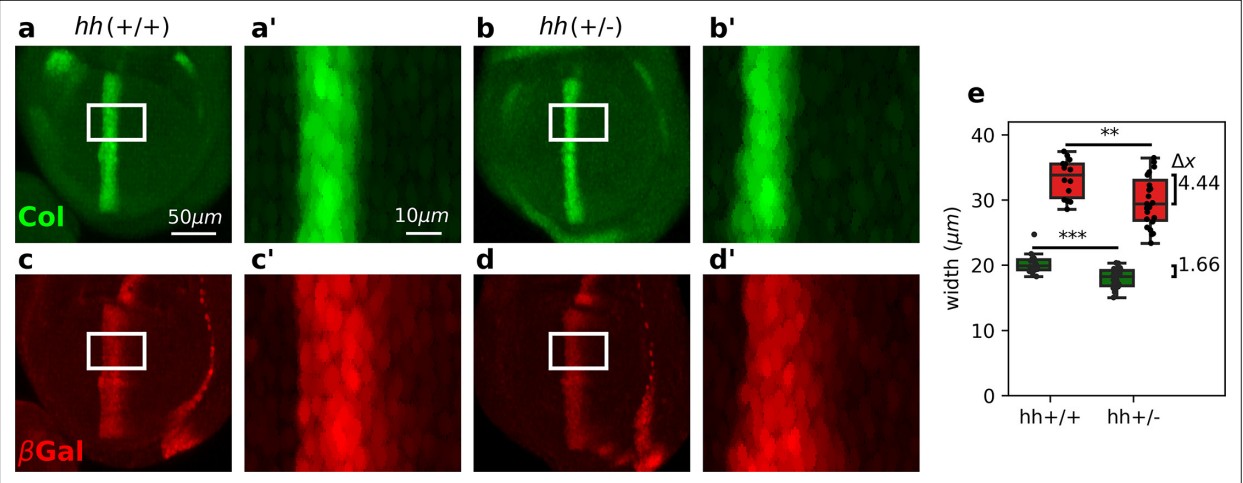

**Figure 3.** Differential robustness of Hh targets to *hh* dosage. (**a-d**) Representative third-instar wild-type, *hh*(+/+) (**a, c**), and *hh* heterozygous *hh*(+/−) (**b, d**) wing discs immunostained with Col (**a, b**) and β-galactosidase (**c, d**) antibodies. Both *hh*(+/+) and *hh*(+/−) flies carry a transgene with a *dpp*LacZ enhancer trap, so β-galactosidase marks the pattern of *dpp* expression. The scale bars in a, a' apply to b, b'; c, c'; and d, d' panels, respectively. (**a'-d'**) Enlarged areas of the white boxes shown in (**a-d**). (**e**) Widths of the *col* and *dpp*LacZ patterns (color coded as in a–d) measured in the region marked by the white rectangle (see *Figure 3—source data 1* and *Figure 3—source code 1*). The brackets on the right represent the difference between the medians of both groups. A non-parametric Mann–Whitney *U* test was applied in both cases (*Figure 3—source data 1*). Statistical p-values are $3.0 \times 10^{-4}$ for Col (**) and $6.0 \times 10^{-3}$ for *dpp*LacZ (***). *hh*(+/−) discs (*n* = 14). *hh*(+/+) discs (*n* = 23). See *Figure 3—source code 1*.

The online version of this article includes the following source data, source code, and figure supplement(s) for figure 3:

**Source code 1.** Code to generate *Figure 3*.

**Source data 1.** Raw data represented in *Figure 3e* .

**Source data 2.** Raw data represented in *Figure 3—figure supplement 1*.

**Figure supplement 1.** The wing disc pouch area does not change in mutant discs.

**Figure supplement 2.** Differences in the width of col and *dpp*LacZ patterns in *hh*(+/+) discs at different threshold values.

---

the overshoot when it reaches a maximum, before it starts to decrease again toward the steady state (*Figure 2c*).

## *col* expression is more robust than *dpp* expression in the *Drosophila* wing disc

We then proceeded to test experimentally whether Hh targets are diferentially robust to changes in Hh dosage as predicted by the overshoot model. Previous studies showed that the width of the *col* domain is largely unaffected in *hh* heterozygous wing discs (*Irons et al., 2010*; *Hatori et al., 2021*). To investigate if this robustness property also holds for *dpp*, which is established by the overshoot (*Nahmad and Stathopoulos, 2009*), we examined the patterns of *col* (using a Col anti-body) and *dpp* (using a *dpp*lacZ reporter) in discs carrying 1 or 2 copies of *hh* (referred as *hh*(+/−) and *hh*(+/+), respectively). We found that the width of the Col pattern in *hh*(+/−) mutant discs is reduced by 1.66 μm relative to *hh*(+/+) wild-type discs (*Figure 3a, b, e*). Although this difference is statistically significant, it is less than the average diameter of a single cell (about 2.5 μm) and therefore, it confirms previous experimental findings (*Irons et al., 2010*; *Hatori et al., 2021*). However, the pattern of *dpp*LacZ is reduced by 4.44 μm in *hh*(+/−) discs relative to *hh*(+/+) controls (*Figure 3c–e*). This result does not depend on the size of the wing disc, since the pouch area in both, *hh*(+/−) and *hh*(+/+) discs are approximately the same (*Figure 3—figure supplement 1*), nor on the threshold used to measure the width of the patterns (see *Figure 3—figure supplement 2*). We conclude that, in agreement with the overshoot model of Hh signaling, but not with any of the steady-state models, the pattern width of Col is more robust than the pattern width of anterior *dpp*LacZ.

## The overshoot model predicts higher precision in the establishment of the *dpp* border than would be expected from the classical steady-state model

Our findings that the width of *dpp* is less robust than the width of *col* in agreement with the overshoot model is puzzling. Why would Hh patterning uses a dynamic mechanism that patterns *dpp* at the time of least robustness (*Figure 2c*)? Why would *col* and *dpp* have different robustness properties (*Figures 1d, 2a, and 3*)? We wondered if this dynamical model trades off one patterning advantage over another in a target-specific manner. Morphogen concentrations are naturally noisy, which may cause territories to have a diffuse border especially when the morphogen narrowly declines due to self-dependent ligand degradation (*Lander et al., 2009*). In particular, we noticed that if *dpp* had to be specified by the steady-state gradient subject to Ptc-dependent degradation, instead that with the overshoot gradient, it would have to be specified at a location where the Hh gradient is nearly flat (*Figure 4a*). But at this same location, the Hh gradient is not as flat (*Figure 4b*). Therefore, we predicted that the overshoot model would establish a more precise *dpp* anterior boundary compared to a steady-state model, suggesting that the dynamic interpretation of Hh signaling would trade off robustness for precision. Therefore, we analyzed the performance of the overshoot and steady-state models at specifying the sharpness of a pattern boundary. We defined a measure of precision, $\sigma_x$, for an experimental or simulated pattern boundary as the standard deviation of different measurements along the extension of the pattern (*Figure 4c*). Evidently, perfect precision occurs for $\sigma_x = 0$, when the pattern would be completely sharp. In contrast, as $\sigma_x$ increases, the less precise the pattern boundary is.

We first measured $\sigma_x$ at the anterior border of *col* and *dpp* in *hh*(+/+) wing discs reported in *Figure 3*. We found that *col* is about twice more precise than *dpp* (*Figure 4d–f*). Then, we compared the precision of the anterior border in simulated patterns of *col* and *dpp* (as defined both by the overshoot and steady-state gradients). To do so, we introduced Gaussian noise in the threshold $T$ at which the Signal function establishes a patterning position (see Materials and methods). Since the mechanism that sets the anterior border of the *col* pattern is the same in both the overshoot and steady-state interpretations, we fitted the extent of noise in the threshold $T$ such that the precision of the simulated border of *col* is the same as the one we measured in the experimental pattern ($\sigma_x$ = 1.23 μm). At this extent of noise in $T$, we compared the simulated border of *dpp* defined by the overshoot ($dpp_{over}$) and steady-state models ($dpp_{SS}$). We found that under the overshoot model, the anterior border of *dpp* is predicted to be more precise than under the steady-state model (*Figure 4f*). Indeed, the overshoot model predicts a sharper border to what is observed experimentally, but this is not biologically significant since $\sigma_x$ is less than one cell diameter in both cases. However, the mean of $\sigma_x$ for the simulated *dpp* border under the steady-state model is 4.36 μm, suggesting that if the anterior border of *dpp* was established by a steady-state gradient, it would have an imprecision of approximately two cell diameters, which could have some patterning impact in the adult wing (see Discussion).

## Discussion

The robust architecture of body plans to genetic and environmental perturbations is a general feature of developmental systems (*Waddington, 1942*; *Csete and Doyle, 2002*; *Kitano, 2004*). At the same time, this robust design should also admit some flexibility in order to allow the system to evolve and adapt under certain genetic or environmental challenges (*Barkai and Shilo, 2007*). While much work has been dedicated to the understanding of network features that confer robustness in developmental patterning, it is unclear how a robust, yet flexible architecture could be encoded in the interpretation of morphogen gradients (*Lander et al., 2009*; *Lo et al., 2015*). In particular, despite much prior theoretical work, the ability of a single morphogen to produce different patterning outputs with target-specific properties has not been studied in detail.

Relative to the classical view of morphogen interpretation, in which different morphogen concentration thresholds at the steady state define different borders of gene expression patterns, two strategies have been proposed to increase robustness to changes in the rates of morphogen production. First, morphogen gradients that promote their own degradation and sharply decay near the source of ligand production (*Eldar et al., 2003*). And second, gradients that specify patterns prior to steady

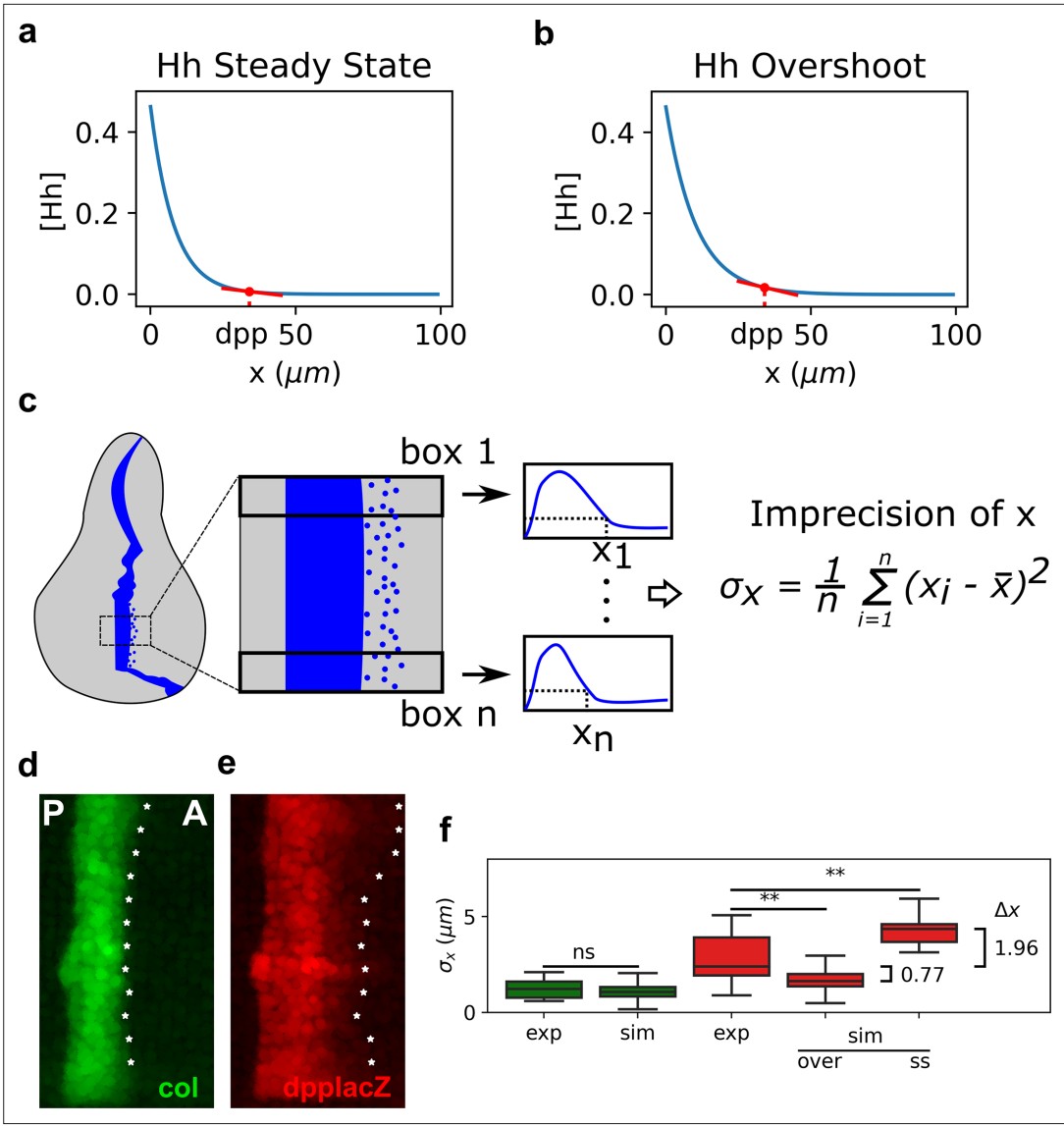

**Figure 4.** The overshoot model predicts more precision of the anterior border of *dpp* than the steady-state model. (a, b). Representation of the steady-state (**a**) and overshoot (**b**) Hh gradients. At the location of the *dpp* anterior border, the slope of the gradient is steeper for the overshoot gradient than for the steady-state gradient. (**c**) Schematic representation of how we define our measure of precision for a patterning border (both in experimental and in simulated patterns). First, a box defines the region of interest (ROI) in the pattern. Then, this ROI is subdivided in $n$ boxes, each of which define a position $x_i$. The measure of precision is the standard deviation of all the $x_i$ values. (d, e) Representative Col (**d**) and *dpp*LacZ (**e**) patterns in which the $x_i$ for each ROI as defined in c is measured and marked with an asterisk along the anterior border. (**f**) Quantification of $\sigma_x$ in several experimental (exp) and simulated (sim) patterns of *col* (green) and *dpp* (red). In the simulated patterns, noise levels are adjusted so that the distributions of *col* are not statistically significant and these noise levels are used to computed the simulated $\sigma_x$ of *dpp* as determined by the steady state (ss) or overshot (over) models. exp sample sizes as in *Figure 3*. sim sample sizes is $n = 50$ in all cases. For the statistical comparison, a Mann–Whitney *U* tests were applied in all cases. Statistical p-value for *col* was $p = 0.42$. For experimental vs. overshoot *dpp*: $p = 1.0 \times 10^{-3}$ (**), and for experimental vs. simulated steady-state *dpp*: $p = 9.0 \times 10^{-3}$ (**).

The online version of this article includes the following source data for figure 4:

**Source code 1.** Code to generate *Figure 4*.

**Source data 1.** Raw data represented in *Figure 4f*.

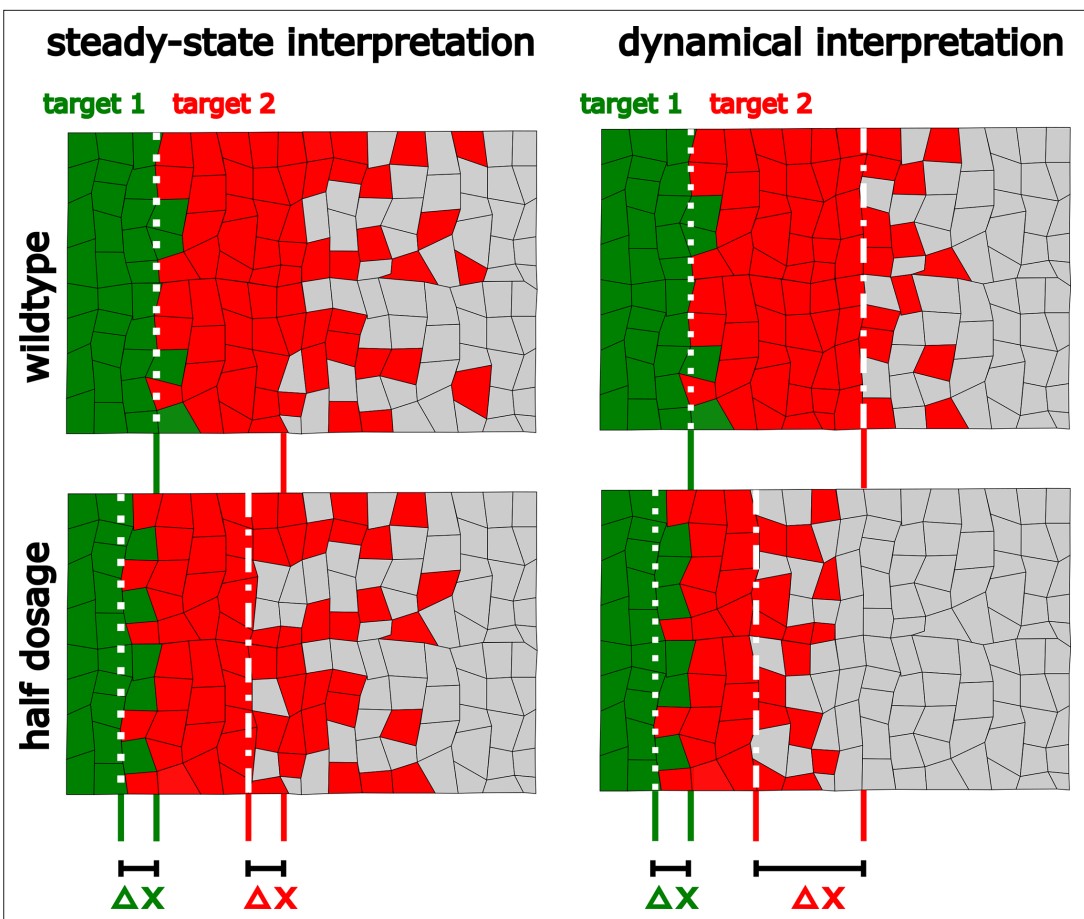

**Figure 5.** The dynamical interpretation of the Hh gradient trades off robustness for higher precision in a target-specific manner. In the steady-state interpretation, all the target genes are established with the same robustness ($\Delta x$) upon perturbations in the amount of ligand. In the overshoot model interpretation one of the target genes (red) is established with less robustness than the other (green). However, it allows the less robust gene to be defined with greater precision than the steady state would define it (compare the sharpness of the boundaries of these patterns).

state (*Bergmann et al., 2007*). When implementing either of these strategies, increased robustness is achieved for all gene expression patterns, regardless of the concentration thresholds at which they are established. However, both of these strategies have a clear inconvenience; they significantly narrow the patterning domain, and therefore, morphogen readout occurs where the gradient is essentially flat (*Adelmann et al., 2023*). Thus, these strategies provide robustness at the expense of a narrower gradient which may result in an imprecise border of gene expression. In agreement with this idea, *Adelmann et al., 2023* recently showed that a linearly decaying gradient establishes more precise patterning boundaries with respect to a gradient established by a self-enhanced ligand degradation mechanism when interpreted several cells away from the morphogen source. The dynamic interpretation of Hh patterning in the *Drosophila* wing disc (*Nahmad and Stathopoulos, 2009*) offers a mechanistic implementation of this idea. First, a linearly decaying Hh gradient (the overshoot gradient) establishes the anterior border of *dpp* prior to upregulation of the Hh receptor, Ptc; once Ptc is upregulated, self-enhanced ligand degradation narrows the gradient and the anterior border of *col* is established (*Figure 1*). Under this model, the *col* border exhibits higher robustness than the *dpp* border to *hh* dosage (*Figure 2*), and our experimental data supports this prediction (*Figure 3*). This reduced robustness of *dpp* patterning occurs as a trade off for increased precision, relative to what would be expected by the steady-state interpretation model (*Figure 4*). Therefore, the dynamical interpretation of Hh signaling offers a target-specific, robust-yet-flexible architecture of patterning in this system (*Figure 5*).

The finding that the displacement of the anterior borders of Hh targets is more than twice for *dpp* than for *col* ($\approx$ 2.65, from their median values; *Figure 3e*) provides a interesting prediction about the overshoot gradient. From *Equation 11*, it can be inferred that the overshoot occurs when Ptc expressions is about twice its basal levels in the anterior compartment, but estimates suggest that Ptc reaches about seven times its basal levels in Ptc domain (*Casali and Struhl, 2004*). This suggests that the overshoot occurs significantly earlier than Ptc reaches its steady-state levels and that Ptc is produced at much larger amounts than what actually is needed to control the range of the Hh gradient. But since unbound Ptc represses Hh signaling, perhaps the purpose of building very high levels of Ptc is to desensitize Hh signaling over time as has been proposed for the vertebrate neural tube (*Dessaud et al., 2008*).

Why does this patterning system is wired to ensure robustness for the *col* border, but favors precision over robustness for *dpp*? In the *Drosophila* wing, the expression of *col* defines directly a specific feature in the adult wing, the L3–L4 intervein area (*Vervoort et al., 1999*), which corresponds to the more central area of the wing, whereas the *dpp* pattern does not have a direct positional role in the adult wing, but it acts as the source of another morphogen. As suggested by prior theoretical work, the source where a morphogen is produced does not have a significant impact on patterning (*Mizutani et al., 2006*), so the robustness of the *dpp* pattern may not subject to strong selection pressure during evolution, or perhaps other mechanisms downstream of Hh signaling exist to provide robustness at the level of Dpp signaling (*Aguilar-Hidalgo et al., 2018*; *Romanova-Michaelides et al., 2022*). In contrast, in the adult wing of *Drosophila*, precision could have a direct role on the sharpness of vein patterning. Thus, robustness ensures the correct positioning of veins whereas precision may be related to ensure straight veins. While it is unclear if a more imprecise *dpp* pattern would impact the straightness of veins 2 and 5 which are positioned by Dpp signaling, it suggests that in general, the overshoot model ensures robust positioning close to the morphogen source, but prioritize straightness of stripe-like patterns over positioning in more distant locations. Given that Ptc-dependent Hh degradation is evolutionary conserved (*Chen and Struhl, 1996*), our findings could have implications for robust and precise patterning in other systems as well.

## Materials and methods

**Key resources table**

| Reagent type (species) or resource | Designation | Source or reference | Identifiers | Additional information |
|---|---|---|---|---|
| strain, strain background (*Drosophila melanogaster*) | *hh*(+/−) allele | Bloomington *Drosophila* Stock Center | 1749 | ry[506] hh[AC]/TM3, Sb. hh[AC] is an amorphic allele |
| strain, strain background (*Drosophila melanogaster*) | *dpp*LacZ | Bloomington *Drosophila* Stock Center | 12379 | cn[1] dpp[10638]/CyO; ry[506]. dpp[10638] is a lacZ is a dpp enhancer trap. |
| antibody | anti-Col (mouse monoclonal) | Gift from M. Crozatier *Vervoort et al., 1999* | | 1:250; overnight incubation |
| antibody | anti-β-gal (rabbit polyclonal) | MP Biomedicals | Cat. # 55976 | 1:250; overnight incubation |
| software, algorithm | Python | this paper | pandas; numpy; OpenCV; matplotlib; seaborn; odeint; solve_bvp | Customized source codes (available from this paper) |

## Fly stocks and crosses

Fly crosses were conducted at 25°C. For experiments using one copy of *hh* [*hh*(+/-)] (*Figure 3*), *ry*[506],*hh*[AC]/TM3,Sb[1] flies (Bloomington *Drosophila Stock Center*, BDSC, #1749) were crossed to *cn*[1],*dpp*10638/CyO (BDSC # 12379) flies at 25°C to obtain *cn*[1],*dpp*[10638]/+; *ry*[506],*hh*[AC]/*ry*[506] discs. *hh*[AC] is a lost of function *hh* allele and *dpp*10638 is a transgene containing a LacZ reporter that drives nuclear β-galactosidase in the location of the *dpp* gene. Control discs with two copies of *hh* [*hh*(+/+)] are obtained from crossing the *dpp*LacZ reporter stock to wild-type flies.

## Wing imaginal disc dissection and immunostaining

Wing imaginal discs were dissected from third-instar larvae. Third-instar larvae were dissected under a stereoscopic microscope and fixed in PEM-T (PEM with 0.1% of Triton X-100) with 4% paraformaldehyde, washed three times, and blocked in PEM-T with 0.5% of bovine serum albumin for 2 hr at room temperature. Then, samples were stained with primary antibodies at 4°C overnight at the following dilutions: monoclonal mouse anti-Col (a gift from M. Crozatier, 1:250), rabbit anti-β-gal (MP Biomedicals, Cat. # 55976, 1:250). Primary antibodies were detected with Alexa Fluor 488 anti-mouse and Alexa Fluor 555 anti-rabbit secondary antibodies (1:1000). Imaging was done in a Leica TC5 SP8 confocal microscope using a 40× oil-immersion objective.

## Numerical simulations

For computations in *Figure 2*, a Forward-in-Time-Centered-in-Space (FTCS) algorithm (using space and time steps of $1\,\mu m$ and time steps of $0.5\,s$, respectively) was implemented to solve *Equations 12–16* in Python, using the parameters reported by *Nahmad and Stathopoulos, 2009*. At the steady state, the equations can be reduced to a single equation in each compartment (*Nahmad and Stathopoulos, 2009*):

$$D\frac{d^2 Hh_{SS}}{dx^2} + S^+(x)\alpha_{Hh} - \frac{\chi S^-(x)Hh_{SS}}{\gamma_{Hh\_Ptc}Hh + \beta_{Ptc}}\left[\alpha_{ptc0} + \frac{\alpha_{ptc}Hh_{SS}^{nm}}{\eta^m\left(k^n + Hh_{SS}^{n\,m}\right) + S^-(x)Hh_{SS}^{nm}}\right] - \beta_{Hh}Hh_{SS} = 0,$$

(18)

where

$$k = \frac{k_{Signal}\beta_{Hh\_Ptc}}{\gamma_{Hh\_Ptc}},$$

$$\eta = \frac{k_{ptc}\beta_{Signal}}{\alpha_{Signal}},$$

$$\chi = \frac{\mu_{Ptc}\gamma_{Hh\_Ptc}}{\beta_{ptc}}.$$

The steady-state *Equation 18* was solved using solve_bvp and solve_ivp from scipy.integrate Python package. Plots were made with matplotlib and seaborn libraries of Python (see *Figure 3— source code 1* and *Figure 4—source code 1*). To compute $\Delta x$ as defined in *Equation 1* in *Figure 2*, we used 0.2 of the maximum value of the *Signal* function and numerically solved for corresponding location $x$.

For simulations of *col* and $dpp_{SS}$ patterns in *Figure 4*, we considered an exponential decay gradient of Hh, like obtained with the simple model in *Figure 1*, evaluated on a matrix of $80 \times 50$. Patterns were determined by the position defined by the threshold $T$ of 20% of the maximum *Signal* value (with a Gaussian noise with mean $T$ and standard deviation determined in such a way that noise of simulated *col* coincides width background distribution noise of experimental *col* pattern, i.e., 1.23 μm). For $dpp_{over}$, we used numerical solution of *Signal* overshoot (i.e., the *Signal* function at the time of maximum range) to fit a Hill function, $A\frac{Hh^n}{k^n + Hh^n}$, using the function fit_curve of scipy.optimize. We found $A = 0.2372$, $k = 0.0483$, and $n = 4.6212$. Then, we used the approximation function of *Signal* overshoot to evaluate an exponential decay gradient of Hh overshoot and made an analysis analogous to what was done at the steady state. We measure the width of the pattern at 0.2 of the profile maximun obtained through a vertical projection of the simulated pattern.

## Image analysis

For image analysis, we took the Z projection of the confocal images using ImageJ. 16-bit resolution images were saved in TIF format and then processed to measure the width of the fluorescence patterns using OpenCv library of Python. We normalized the intensity values after dividing them by the maximum intensity value and then we measured the width of each pattern domain at 0.2 of relative intensity (in *Figure 3—figure supplement 2* we varied this threshold value from 0.1 to 0.6). Graphs were plotted with matplotlib and seaborn libraries of Python (see Source code for each panel of *Figure 2*). The same images were used to measure robustness (*Figure 3*) and precision (*Figure 4*).

## Acknowledgements

We thank M Crozatier for kindly providing us with an aliquot of Collier antibody. We also thank Fanis Misirlis, José Antonio Arias, and members of the Nahmad laboratory for useful discussions, and Rafael Rodríguez-Muñoz and José Luis Fernández for technical assistance.

## Additional information

### Competing interests

Marcos Nahmad: Reviewing editor, eLife. The other authors declare that no competing interests exist.

### Funding

| Funder | Grant reference number | Author |
|---|---|---|
| Centre for Research and Advanced Studies of the National Polytechnic Institute, Mexico | Institutional Support | Marcos Nahmad |
| Consejo Nacional de Humanidades, Ciencias y Tecnologías | Graduate Fellowship | Rosalío Reyes |
| National Institutes of Health | GM076516 | Arthur D Lander |

The funders had no role in study design, data collection, and interpretation, or the decision to submit the work for publication.

### Author contributions

Rosalío Reyes, Conceptualization, Resources, Formal analysis, Validation, Investigation, Visualization, Methodology, Writing – original draft; Arthur D Lander, Conceptualization, Investigation, Methodology, Project administration, Writing – review and editing; Marcos Nahmad, Conceptualization, Formal analysis, Supervision, Validation, Investigation, Methodology, Writing – original draft, Project administration

### Author ORCIDs

Arthur D Lander (b) https://orcid.org/0000-0002-4380-5525
Marcos Nahmad (b) https://orcid.org/0000-0001-6300-5608

### Decision letter and Author response

Decision letter https://doi.org/10.7554/eLife.85755.sa1
Author response https://doi.org/10.7554/eLife.85755.sa2

## Additional files

### Supplementary files

• MDAR checklist

### Data availability

All data generated or analyzed during this study (including the source code) are included in this submission.

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
