## [Editor Report]

This study presents a valuable finding on the precision conferred by dynamical interpretation of morphogen gradients. The evidence supporting the claims of the authors is convincing, with compelling theoretical analysis and solid experimental data. The authors have adequately addressed most concerns raised and so the work will be of considerable interest to the developmental biology and developmental systems biology communities.

---

## [Decision Letter]

**Decision letter after peer review:**

Thank you for submitting your article "Dynamic readout of the Hh gradient in the *Drosophila* wing disc reveals pattern-specific tradeoffs between robustness and precision" for consideration by *eLife*. Your article has been reviewed by 2 peer reviewers, and the evaluation has been overseen by David James as the Senior and Reviewing Editor.

Essential revisions:

While your manuscript was deemed of interest there were significant shortcomings identified that need to be addressed. Most notably both referees felt the experimental part was less compelling than the modelling part and in fact, one referee indicated that they felt this was at best an incremental advance over previous findings. We would like to provide you with an opportunity to address these serious concerns as both reviewers did see positive aspects of the study. However, it is critical that you address the issue concerning the nature of the advance compared to previous studies before we can proceed.

The reviewer found this study presents at best incremental advances to the field. It doesn't provide substantial progress conceptually or experimentally from Eldar et al., 2003, Adleman et al., 2022 and particularly Nahmad and Stathopoulos, 2009. The experimental data and interpretation appear to lack the rigor needed to challenge the model predictions.

*Reviewer #1 (Recommendations for the authors):*

The manuscript presents an elegant theoretical analysis of robustness and precision in morphogen ingredients, focusing on hedgehog signaling. I have found the proposal made by the authors interesting and convincing. However, I have found that some parts of the manuscript are not very clear. In addition, I believe the experimental results need to be improved in their presentation and to be broadened in scope if possible. Here below I detail my comments:

1) In the Introduction, paragraph starting at 75 indicates the properties of Hh signaling as if they were disconnected to the features described in the previous paragraph. Please, rewrite it to make all appropriate connections with the previous paragraph.

2) Clarify how robustness is exactly defined. The displacement of the boundary of the pattern upon perturbation of Hh level is used in Figure 1 to say whether a target is more robust. However, the coefficient of robustness is not defined as such displacement. These different definitions should be related and preferably refer to them with different names. In addition, the meaning of m in the definition of the coefficient of robustness is not totally clear to me. A plot depicting it would help. Is m the slope of the non-perturbed gradient at the threshold?

3) The coefficient of robustness used is a different measure of the Robustness introduced by Eldar et al.2003. The latter one considered the displacement upon perturbation relative to the extent of the unperturbed gradient. Why the authors do not use the definition of robustness introduced by Eldar et al? Why the definition of robustness in this manuscript does not take into account whether the gradient spans over a larger or a smaller spatial region? The overshoot gradient produces larger displacements yet it is a gradient spanning a larger domain than the steady-state gradient. I am not sure whether the over-shoot gradient is less robust than the steady gradient if the definition of robustness introduced by Eldar et al. 2003 is used. Please justify and clarify all this.

4) These differences in definitions (point 3) make the comparison of the analysis in Box2 with the results from Eldar et al.2003, described in lines 168-169, awkward. Box 2 analyses exponential gradients. It compares the robustness of two exponential gradients with different spatial characteristic lengths (λ). Based on the definition of the coefficient of robustness of this manuscript, these two exponential gradients have a different robustness. However, if we use the definition of robustness by Eldar et al. 2003, all exponential gradients have the same robustness, R=1, independently of their characteristic length λ. Please clarify.

5) In the text, at the beginning of section 2.3, state more explicitly the concept of precision.

6) Define mathematically how precision is measured. The text refers to Box2 (line 187) but there is no definition of coefficient of precision in that Box (nowhere else either).

7) As far as I understand, precision is related to how fluctuations (noise) on the amount of morphogen impact on the position of the boundary. These fluctuations can be from cell to cell and over time within the same cell. The current manuscript does not model fluctuations or noise. Instead, it uses the slope of the deterministic gradient to define the precision (lines 188-190, using Figure 2A to visualize this idea). The manuscript would benefit from indicating the assumptions behind this claim :

A) It assumes uniform noise, i.e. that noise/fluctuations are independent of the slope of the gradient, in other words, are of the same amplitude at any spatial position. Indeed, what we may expect is not this, since intrinsic noise is proportional to the square root of the number of molecules. Hence, the fluctuations will be larger where the morphogen is in high amounts than where it is in low amounts.

B) It also assumes that the range of Hh concentrations that are not discernible/distinguishable under fluctuations (i.e the widths of the red and green bands in the Hh axis) is independent of the Hh concentration (i.e the width of the red band is located around Hh=0.1 and has the same width as that of the green band which is located at Hh=0.77), and that this range does not change over time (it is the same for the steady and the overshoot gradients).

8) The "Dynamical interpretation" model is used with two (related) different meanings, in my opinion, and this drives confusion. On the one hand, according to Figure 1A',B',C', the Dynamical interpretation model corresponds to a single threshold used by different targets: one uses it in the steady gradient and the other target uses it in the overshoot gradient. On the other hand, in the text, in line 198, the dynamic interpretation is used only to refer to the overshoot gradient. I suggest revising how "dynamical interpretation" is used: whether it applies only to the overshoot gradient and then whether a different name must be used to the whole framework of single-threshold interpretation.

9) The results assume that Dpp and col use the same threshold. This is supported by Nahmad and Stathopoulos 2009. Which threshold value is used? Which value is used for the simulations with different sets of the parameter values?

10) Why Robustness is not analysed for the Signal (x)? I would expect that the target is activated by the Signal and not directly by the morphogen gradient. Hence it is valuable to analyse the robustness in the signal and to add these results. Perhaps Figure 3A-C already compute the magnitudes from the signal profile (and not from the morphogen Hh(x) profile), but it is unclear from the main text and figure caption.

11) In Figure 3 precision is much less analyzed than robustness. I suggest that the type of analysis already done in Figure 3B and C for robustness is also done for precision. These analyses will show whether the conclusions on precision are maintained for different parameter values. By the way, "parameters are varied between 0,5 and 2 of the reported values" means that they are varied between 0,5 and 2 TIMES the reported values? Perhaps is standard but the meaning of the sentence was unclear to me.

12) How the overshoot gradient is identified for the different set of parameters to compute Figure 3B?

13) I suggest computing Figure 4B for the overshoot gradient and therefore show that the trend in Figure 4A is kept for different parameter values.

14) Figures 5-6 should be improved by adding: Scale bars, magnifications of images, and detail at cell resolution to observe the displacements in terms of cell length scales. What is exactly measured should be also depicted: How the width is measured and which width is measured for the blurry boundary of Dpp? Which is the number of samples?

15) The finding that the robustness of Col depends on Ptc regulation supports the results by Eldar et al. 2003 and that Col is a target of the steady gradient. Hence these new experimental results support proposals made in previous papers. In my opinion, this experimental result in this manuscript (section 2.7) is not very relevant since it validates previous proposals but not the new ones from this manuscript.

16) The manuscript indicates that Dpp is less robust but more precise than it would be if it was specified by the steady-state gradient. Since the authors have analysed the case of non-regulated patch, I suggest addressing how Dpp would change when patched is not regulated, and to address it both theoretically, and if possible, experimentally. If Patched is not regulated, then there will not be an overshoot gradient and Dpp should be as robust as col. Is this indeed the theoretical prediction? And experimentally: what is observed? In addition, will precision become worse or better? What is the prediction from the model when patch is not regulated?

*Reviewer #2 (Recommendations for the authors):*

Figure 5 – elaborate on how exactly the results are consistent with the model predictions? While the Dpp width changes more, the width is also larger to begin with- taking into account these rather small changes, can a much simpler model with noise explain the experimental results already (does one have to resort to overshoot and dynamic interpretation?)

Width panels: individual data points should be shown, with "n" defined in the legends

---

## [Author Response]

Essential revisions:While your manuscript was deemed of interest there were significant shortcomings identified that need to be addressed. Most notably both referees felt the experimental part was less compelling than the modelling part and in fact, one referee indicated that they felt this was at best an incremental advance over previous findings. We would like to provide you with an opportunity to address these serious concerns as both reviewers did see positive aspects of the study. However, it is critical that you address the issue concerning the nature of the advance compared to previous studies before we can proceed.The reviewer found this study presents at best incremental advances to the field. It doesn't provide substantial progress conceptually or experimentally from Eldar et al., 2003, Adleman et al., 2022 and particularly Nahmad and Stathopoulos, 2009. The experimental data and interpretation appear to lack the rigor needed to challenge the model predictions.

We truly appreciate the summary and the valuable criticisms that the reviewers raised about of our work. We think that the reviewers’ comments have made the current, major-revised manuscript a much better paper, so we very grateful for their feedback.

First of all, we would like to admit that in the original manuscript our definitions of precision and robustness were confusing and we agree that there is not a consensus about these concepts in the literature. In the revised version of the manuscript, we have independently defined the positional buffering effects as a result of *hh* dosage, as robustness, and sharpness of the anterior borders of patterns, as precision. We also clarified our image analysis of the *col* and *dpp* patterns in agreement with the reviewers’ suggestions. When we used our new definitions to compute robustness and precision on both experimental and simulated patterns, we confirmed the findings of our original manuscript, namely, that the anterior border of *col* that depends on the steady-state Hh gradient subject to self-enhanced degradation gradient is more robust than that of *dpp,* which is set by the overshoot gradient. Nonetheless, the lack of robustness of the *dpp* anterior border with the dynamic gradient is compensated with more precision than would be expected from the steady-state gradient interpretation at this position. In our revised manuscript, we provide experimental evidence for differential robustness to *hh* dosage and use simulations to support our precision hypothesis. Taken together, our work stands out from previous work such as Eldar et al. 2003, Adelmann et al. 2023, and Nahmad and Stathopoulos 2009, to show for the first time that a dynamic interpretation through the overshoot model modulates positioning and sharpness in a target-specific manner during Hh patterning in the *Drosophila* wing disc.

Reviewer #1 (Recommendations for the authors):The manuscript presents an elegant theoretical analysis of robustness and precision in morphogen ingredients, focusing on hedgehog signaling. I have found the proposal made by the authors interesting and convincing. However, I have found that some parts of the manuscript are not very clear. In addition, I believe the experimental results need to be improved in their presentation and to be broadened in scope if possible. Here below I detail my comments:1) In the Introduction, paragraph starting at 75 indicates the properties of Hh signaling as if they were disconnected to the features described in the previous paragraph. Please, rewrite it to make all appropriate connections with the previous paragraph.

Thank you for this comment. We have re-written the introduction to integrate these paragraphs.

2) Clarify how robustness is exactly defined. The displacement of the boundary of the pattern upon perturbation of Hh level is used in Figure 1 to say whether a target is more robust. However, the coefficient of robustness is not defined as such displacement. These different definitions should be related and preferably refer to them with different names. In addition, the meaning of m in the definition of the coefficient of robustness is not totally clear to me. A plot depicting it would help. Is m the slope of the non-perturbed gradient at the threshold?

We appreciate this criticism. We agree that the use of the coefficient of robustness was not necessary and could lead to unnecessary confusion. Therefore, in the revised manuscript, we completely removed the use of the robustness coefficient and instead, we use the very intuitive notion of pattern displacement as a measure of robustness (equation 1 in the revised manuscript). Note that for simple models of Hh signaling such as the one now depicted in Figure 1, the displacement resulting from the steady state and the overshoot gradients can be compared and our claim that steady-state outputs are more robust than overshoot outputs can be directly demostrated (equation 2 in the revised manuscript).

3) The coefficient of robustness used is a different measure of the Robustness introduced by Eldar et al.2003. The latter one considered the displacement upon perturbation relative to the extent of the unperturbed gradient. Why the authors do not use the definition of robustness introduced by Eldar et al? Why the definition of robustness in this manuscript does not take into account whether the gradient spans over a larger or a smaller spatial region? The overshoot gradient produces larger displacements yet it is a gradient spanning a larger domain than the steady-state gradient. I am not sure whether the over-shoot gradient is less robust than the steady gradient if the definition of robustness introduced by Eldar et al. 2003 is used. Please justify and clarify all this.

As we mentioned in the previous point, we no longer use a definition of robustness coefficient, as this is not necessary to address our hypothesis. While the definition of robustness coefficient introduced by Eldar et al. and others allows generalizing the notion of robustness, we think that simply using the displacement between a perturbed and unperturbed location established by a morphogen (equation 1) is a more intuitive measure of robustness that allows to directly show that in fact the outputs of the overshoot gradient are less robust than those of the steady-state gradient (equation 10). Indeed, Eldar et al. 2003 also use this displacement to show the differences in robustness between linear and non-linear gradients (see Eldar et al. 2003, equations 2-5).

4) These differences in definitions (point 3) make the comparison of the analysis in Box2 with the results from Eldar et al.2003, described in lines 168-169, awkward. Box 2 analyses exponential gradients. It compares the robustness of two exponential gradients with different spatial characteristic lengths (λ). Based on the definition of the coefficient of robustness of this manuscript, these two exponential gradients have a different robustness. However, if we use the definition of robustness by Eldar et al. 2003, all exponential gradients have the same robustness, R=1, independently of their characteristic length λ. Please clarify.

We agree that the use of boxes was not a very convenient way of presenting the information. In the revised manuscript, we removed all boxes and put all relevant information directly in the text. Exponential gradients that are perturbed at the boundary conditions do display a shift that depends on the characteristic length λ (as shown also by Eldar et al. 2003, equation 2). Using this displacement as a direct measure of robustness, a larger displacement (i.e., less robustness) occurs in the overshoot gradient that has a larger λ.

5) In the text, at the beginning of section 2.3, state more explicitly the concept of precision.

We have introduced a more direct notion of precision in terms of sharpness of a 2D border (Figure 4c). We think that this new notion of precision is very easy to compute directly from experimental or simulated 2D patterns and effectively reflects the sharpness of a border (see last subsection of the Results).

6) Define mathematically how precision is measured. The text refers to Box2 (line 187) but there is no definition of coefficient of precision in that Box (nowhere else either).

Sorry for being unclear about this in the original manuscript. In the revised manuscript, we completely separated the notions of robustness and precision, so that they can be independently evaluated. Similarly as with robustness, we no longer rely on the coefficient of precision. As stated in our previous point, a mathematical definition of precision is provided in the last subsection of the Results and in Figure 4c.

7) As far as I understand, precision is related to how fluctuations (noise) on the amount of morphogen impact on the position of the boundary. These fluctuations can be from cell to cell and over time within the same cell. The current manuscript does not model fluctuations or noise. Instead, it uses the slope of the deterministic gradient to define the precision (lines 188-190, using Figure 2A to visualize this idea). The manuscript would benefit from indicating the assumptions behind this claim :A) It assumes uniform noise, i.e. that noise/fluctuations are independent of the slope of the gradient, in other words, are of the same amplitude at any spatial position. Indeed, what we may expect is not this, since intrinsic noise is proportional to the square root of the number of molecules. Hence, the fluctuations will be larger where the morphogen is in high amounts than where it is in low amounts.B) It also assumes that the range of Hh concentrations that are not discernible/distinguishable under fluctuations (i.e the widths of the red and green bands in the Hh axis) is independent of the Hh concentration (i.e the width of the red band is located around Hh=0.1 and has the same width as that of the green band which is located at Hh=0.77), and that this range does not change over time (it is the same for the steady and the overshoot gradients).

We appreciate the reviewer pointed this out as it helped us to redefine our notion of precision. It is true that precision is ill-defined in the literature and therefore people refer to precision in different ways, so it is really important to set a clear definition of precision. In her/his comment, the reviewer refers to precision as a change in a patterning position due to fluctuations in morphogen amounts. This notion, applied to a 2D border results in local changes in position along the pattern (Figure 4c), so that effectively it is a measure of sharpness of the patterning border. We no longer need to clarify which of the assumptions referred by the reviewer hold because we are no longer defining precision in terms of the slope of a deterministic gradient. Thanks to this reviewer’s comment, we approached the notion of precision in a different way than in the original manuscript. Namely, we generated simulated data by directly introducing Gaussian noise to the morphogen threshold at which cell fate takes place; we then fixed these noise levels so that simulated data fits the experimental data for the anterior border of *col* (which was not the focus of the precision analysis), and finally, we tested our hypothesis for the anterior border of the *dpp* pattern (see Materials of Methods in the revised manuscript).

8) The "Dynamical interpretation" model is used with two (related) different meanings, in my opinion, and this drives confusion. On the one hand, according to Figure 1A',B',C', the Dynamical interpretation model corresponds to a single threshold used by different targets: one uses it in the steady gradient and the other target uses it in the overshoot gradient. On the other hand, in the text, in line 198, the dynamic interpretation is used only to refer to the overshoot gradient. I suggest revising how "dynamical interpretation" is used: whether it applies only to the overshoot gradient and then whether a different name must be used to the whole framework of single-threshold interpretation.

We understand and apologize for the confusion. In response to this comment, in the revised manuscript we use the term overshoot model every time we refer to the interpretation of Hh signaling (both with the overshoot and steady-state gradient), and we only use dynamical interpretation when referring more generally to the use of dynamic properties of a morphogen gradient.

9) The results assume that Dpp and col use the same threshold. This is supported by Nahmad and Stathopoulos 2009. Which threshold value is used? Which value is used for the simulations with different sets of the parameter values?

We always use 0.2 of the maximum value as the threshold value not only for the simulated data, but also as a threshold to determine pattern boundaries (see Materials and methods’ sections 4.3 and 4.4). It is important to state, however, that our main robustness result that steady-state outputs are more robust than overshoot outputs is independent of the threshold used.

10) Why Robustness is not analysed for the Signal (x)? I would expect that the target is activated by the Signal and not directly by the morphogen gradient. Hence it is valuable to analyse the robustness in the signal and to add these results. Perhaps Figure 3A-C already compute the magnitudes from the signal profile (and not from the morphogen Hh(x) profile), but it is unclear from the main text and figure caption.

We did use Signal when computed x in Figure 2 of the revised manuscript. Thank you for the observation. We are now stating this in the main text as well as in the legend of Figure 2.

11) In Figure 3 precision is much less analyzed than robustness. I suggest that the type of analysis already done in Figure 3B and C for robustness is also done for precision. These analyses will show whether the conclusions on precision are maintained for different parameter values. By the way, "parameters are varied between 0,5 and 2 of the reported values" means that they are varied between 0,5 and 2 TIMES the reported values? Perhaps is standard but the meaning of the sentence was unclear to me.

In the revised manuscript, robustness and precision are analyzed independently, so I don’t think that the first part of the comment applies. Please keep in mind, that our analysis of robustness is limited to changes in hh dosage, but for precision, we simply use Gaussian noise in the interpretation of the gradient. Thus, the analysis for robustness presented in Figure 2 of the revised manuscript does not apply to precision and the analysis of precision presented in Figure 4 of the revised manuscript does not apply to robustness. Regarding the second part of the comment; yes, it refers to 0.5 to 2 TIMES and this is now clearly specified in Figure 2a,b.

12) How the overshoot gradient is identified for the different set of parameters to compute Figure 3B?

The overshoot is defined in the main text as ‘the transient gradient of maximum range’. In our simulations of Figure 2 in the revised manuscript, this is how overshoot displacements are computed.

13) I suggest computing Figure 4B for the overshoot gradient and therefore show that the trend in Figure 4A is kept for different parameter values.

Done. This is shown in new Figure 2b’.

14) Figures 5-6 should be improved by adding: Scale bars, magnifications of images, and detail at cell resolution to observe the displacements in terms of cell length scales. What is exactly measured should be also depicted: How the width is measured and which width is measured for the blurry boundary of Dpp? Which is the number of samples?

Thank you for the suggestions. In the new manuscript, we have updated new Figure 3 (which corresponds to Figure 5 in the original manuscript; note that Figure 6 from the original manuscript is no longer in the revision) incorporating these suggestions. Information on the number of samples and how widths are measured is clearly explained in the legend of Figure 3 and in Materials and methods section.

15) The finding that the robustness of Col depends on Ptc regulation supports the results by Eldar et al. 2003 and that Col is a target of the steady gradient. Hence these new experimental results support proposals made in previous papers. In my opinion, this experimental result in this manuscript (section 2.7) is not very relevant since it validates previous proposals but not the new ones from this manuscript.

We agree with the reviewer that Figure 6 did not add anything new to the manuscript and it was difficult to fit in the story. In addition, it was difficult to interpret experimentally due to the effects in changes in temperature (a point that was also raised the other reviewer). Therefore, we decided to remove it from the revised manuscript. This decision of course does not affect the conclusions of the paper and makes it a more concise story. We thank the reviewer for this suggestion.

16) The manuscript indicates that Dpp is less robust but more precise than it would be if it was specified by the steady-state gradient. Since the authors have analysed the case of non-regulated patch, I suggest addressing how Dpp would change when patched is not regulated, and to address it both theoretically, and if possible, experimentally. If Patched is not regulated, then there will not be an overshoot gradient and Dpp should be as robust as col. Is this indeed the theoretical prediction? And experimentally: what is observed? In addition, will precision become worse or better? What is the prediction from the model when patch is not regulated?

As the reviewer rightly pointed out, when *ptc* is not upregulated, there is no overshoot and therefore, the anterior border of *col* and *dpp* overlap. Indeed, Nahmad and Stathopoulos (2009) already showed that this is the case experimentally. The prediction of the model, with no overshoot is that the patterns would be identical (simply, because by definition of overshoot as the gradient with maximum range, the overshoot and the steady-state gradients would be the same) and therefore their robustness and precision would be also identical. However, it is possible to compare, both in the simulations and experimentally, the precision of the anterior borders of *col* and *dpp*. In the revised manuscript, we computed the precision of *col* and *dpp* experimentally using the same protocol and showed that the anterior border of *col* is more precise than that of *dpp* which confirms the observation by eye that the *dpp* border is more fuzzy than the *col* border (Figure 3). Moreover, using this measure of precision, we are able to show that the anterior border of *dpp* in simulations is more precise when obtained it from the overshoot gradient than from the steady-state gradient.

Reviewer #2 (Recommendations for the authors):Figure 5 – elaborate on how exactly the results are consistent with the model predictions? While the Dpp width changes more, the width is also larger to begin with- taking into account these rather small changes, can a much simpler model with noise explain the experimental results already (does one have to resort to overshoot and dynamic interpretation?)

We did not make measurements relative to the width of each pattern because robustness (as a displacement) and precision (sharpness) should be interpreted in absolute units. In our manuscript, we show that simply adding noise to a steady-state model could not explain differential robustness. We cannot rule out that other models could also explain differential robustness, but we are favoring our interpretation on a model that has been validated by prior experimental work (i.e., the overshoot model; Nahmad and Stathopoulos, 2009).

Width panels: individual data points should be shown, with "n" defined in the legends

We now include individual data points in Figure 3 of the revised manuscript and display the sample numbers ‘n’ in the figure legend.